# Medial Meniscus Posterior Root Tear: How Far Have We Come and What Remains?

**DOI:** 10.3390/medicina59071181

**Published:** 2023-06-21

**Authors:** Hyun-Soo Moon, Chong-Hyuk Choi, Min Jung, Kwangho Chung, Se-Han Jung, Yun-Hyeok Kim, Sung-Hwan Kim

**Affiliations:** 1Arthroscopy and Joint Research Institute, Yonsei University College of Medicine, Seoul 03722, Republic of Korea; 2Department of Orthopedic Surgery, Hallym University Sacred Heart Hospital, Hallym University College of Medicine, Anyang 14068, Republic of Korea; 3Department of Orthopedic Surgery, Severance Hospital, Yonsei University College of Medicine, Seoul 03722, Republic of Korea; 4Department of Orthopedic Surgery, Yongin Severance Hospital, Yonsei University College of Medicine, Yongin 16995, Republic of Korea; 5Department of Orthopedic Surgery, Gangnam Severance Hospital, Yonsei University College of Medicine, Seoul 06273, Republic of Korea

**Keywords:** meniscus root, medial meniscus posterior root, root tear, medial meniscus posterior root tear, meniscus root repair, transtibial pull-out repair

## Abstract

Medial meniscus posterior root tears (MMRTs), defined as tears or avulsions that occur within 1 cm of the tibial attachment of the medial meniscus posterior root, lead to biomechanically detrimental knee conditions by creating a functionally meniscal-deficient status. Given their biomechanical significance, MMRTs have recently been gaining increasing interest. Accordingly, numerous studies have been conducted on the anatomy, biomechanics, clinical features, diagnosis, and treatment of MMRTs, and extensive knowledge has been accumulated. Although a consensus has not yet been reached on several issues, such as surgical indications, surgical techniques, and rehabilitation protocols, this article aimed to comprehensively review the current knowledge on MMRTs and to introduce the author’s treatment strategies.

## 1. Introduction

Over the past few decades, perceptions of the clinical importance of menisci have changed [1]. The meniscus was once considered a functionless remnant vestige [2]; however, as a result of extensive research over a long time, it is now accepted as one of the most important structures of the knee joint. The meniscus is responsible for approximately 40–80% of the load transmission in the knee joint [3], and damage to this structure increases the peak contact pressure and decreases the contact area of the articular surface of the knee [4]. In addition, the meniscus plays several vital biomechanical roles, such as shock absorption, joint stabilization, lubrication, and proprioception [5]. Accordingly, a paradigm of treatment for meniscus tears has changed from resection to preservation [1,6].

There are various types of meniscus tears based on their location and patterns, each with different characteristics [7,8,9]. Among these, a medial meniscus posterior root tear (MMRT) is considered a detrimental injury because it causes conditions similar to those of total meniscectomy [10]. Although an MMRT was not well known until approximately a decade ago, numerous studies conducted with increasing attention to its biomechanical and clinical importance have been reported [11]. MMRTs account for approximately 10% of all types of meniscus tears and 22–28% of medial meniscus tears [12,13,14,15,16]. Given that MMRTs commonly precede advanced knee osteoarthritis [17], the actual incidence is considered higher than the reported one. Although much remains to be studied and understood, the multifaceted findings on MMRTs currently form the basis of their treatment strategies. Several review papers have been published on meniscus root tears; however, only a few articles have provided a comprehensive summary of the latest knowledge specifically focusing on MMRTs. Therefore, we aimed to comprehensively review the current knowledge on MMRTs, including their anatomy, biomechanics, clinical presentation, treatment options, and clinical outcomes, and we provide our treatment strategies.

## 2. Medial Meniscus Posterior Root Tear

### 2.1. Anatomy

The medial meniscus posterior root (MMPR), the tibial attachment of the medial meniscus posterior horn, is a ligamentous structure that anchors the meniscus to the tibia. The structural transition from a fibrocartilaginous meniscus body to a ligament-like structure facilitates the load transmission from the meniscus to the bone [18]. The MMPR is mainly composed of collagen fibers running parallel to the longitudinal axis of the meniscus, which also contributes to resisting the tensile force applied to the meniscus [19]. Owing to these mechanical characteristics, the MMPR stabilizes the meniscus between the femoral condyle and tibial plateau and allows effective load transmission [10]. The MMPR is located approximately 9.6 mm posterior and 0.7 mm lateral from the medial tibial eminence apex, with an area of the attachment site of approximately 30.4 mm^2^ [20] (Figure 1A). In addition, the MMPR has diagonally-oriented posterior fibrous expansions called shiny white fibers, which increase the attachment area of the MMPR to the tibial plateau [20,21]. Shiny white fibers are used as anatomical landmarks during surgical procedures, involving the MMPR or posterior cruciate ligament [21,22].

### 2.2. Definition and Classification

An MMRT is defined as a tear or avulsion that occurs within 1 cm of the tibial attachment of the MMPR [23] (Figure 1B). This type of meniscal injury can be classified according to the system by LaPrade et al. as follows: (1) type 1, partial stable tear; (2) type 2, complete radial tear; (3) type 3, bucket-handle tear with meniscus root detachment; (4) type 4, complex oblique tear extending into the root attachment; and (5) type 5, avulsion fracture of the root attachment [23]. This concise classification system is clinically beneficial because it provides criteria for distinguishing MMRTs from other types of meniscus tears and helps determine treatment options. Furthermore, this classification system provides potential criteria for predicting the chronicity of an MMRT. Types 1, 2, and 4 are associated with acute or chronic tears, whereas types 3 and 5 are usually found in acute traumatic tears [23,24]. Among the five types of MMRTs, type 2 is the most common [23,24].

## 3. Biomechanics

The MMPR plays an important role in transmitting and distributing the load applied to the meniscus, thereby preserving the adjacent articular cartilage. The hoop tension generated by the collagen fibers that make up the meniscus and its root attachment counteracts the compressive forces applied to the knee joint while maintaining the meniscus between the femur and tibia [5]. A radial tear of these structures disrupts hoop tension. This disruption pushes the meniscus out between the two bones, eventually causing the meniscus to lose its function (Figure 1B). In a cadaveric study, Allaire et al. reported that an MMRT increased the contact pressure in the medial compartment of the knee by 25%, equivalent to that after total medial meniscectomy [10]. Subsequent biomechanical studies have consistently shown that an MMRT increases contact pressure while decreasing the contact area in the medial compartment of the knee [25,26,27]. On the other hand, the repair of an MMRT restores joint mechanics to a condition similar to that of an intact knee under time-zero conditions [10,25,26,27]. However, the non-anatomical repair may not restore the native biomechanics of the knee [28]. In addition to loading profiles, MMRTs have been reported to directly affect knee stability by causing rotation and translation of the joint [10].

## 4. Clinical Presentation

### 4.1. Pathophysiology and Clinical Features

MMRTs are caused by acute traumatic events or degenerative changes [29]. Although there are no clear criteria for differentiating them, degenerative MMRTs are thought to occur during light daily activities or without accompanying acute injuries to other knee structures.

Most traumatic MMRTs occur in association with ligament injuries of the knee. Anterior cruciate ligament injuries can accompany MMRTs [30]. In particular, MMRTs can be observed in cases of multi-ligament injuries [31,32,33], and varus injury patterns may further increase the possibility of MMRTs [32]. The incidence of traumatic MMRTs associated with ligament injuries of the knee ranges from 3% to 14% [31,32,33].

Degenerative MMRTs are relatively more common than traumatic MMRTs and are more susceptible to tears than root tears in other locations [29]. The medial meniscus posterior horn carries a large portion of the load applied to the knee joint, including compressive and shear forces [3]. Furthermore, the MMPR shows the lowest mobility of all meniscus roots, making it vulnerable to degenerative tears [34]. Park et al. reported that the degree of MMPR degeneration was highly associated with the degree of an MMRT and that degeneration, accompanied by fibrocartilage metaplasia or calcification, may precede an MMRT [35]. Possible risk factors for these types of MMRTs include older age, female sex, increased body mass index (BMI), increased varus alignment of the lower extremity, and decreased sports activity levels [14]. MMRTs tend to occur during light daily activities, such as squatting, rising from a chair, or going up or down stairs [12,36]. Interestingly, compared with other meniscus tears, MMRTs showed a relatively clear onset of symptoms. The onset of acute symptoms is commonly expressed as painful popping [13], and approximately 85–91% of patients state that they clearly recall the time of symptom onset [37,38]. According to Bae et al., a single painful popping sensation could be a highly predictive clinical sign for identifying MMRTs, with a positive predictive value of 96.5% [13]. In terms of physical examination, several parameters, such as pain on full flexion, joint line tenderness, and the McMurray test, can be used; however, there are no specific physical findings to diagnose MMRTs [12,39]. The Akmese sign, with severe medial joint line tenderness in near extension and minimal or no tenderness in knee hyperflexion, has recently been proposed as a possible physical examination to distinguish MMRTs from other medial meniscus pathologies, with high sensitivity and specificity [40].

### 4.2. Imaging

Magnetic resonance imaging (MRI) is the gold standard for diagnosing MMRTs. Although the quality of imaging and the ability of the observer may affect the accuracy of image interpretation, MRI is known to have a high accuracy in identifying MMRTs. According to Lee et al., the sensitivity, specificity, and accuracy of MRI for the detection of MMRTs are 86–90%, 94–95%, and 94%, respectively [41]. MMRTs show characteristic radiographic findings in each plane on MRI, which are as follows: (1) vertical high signal at the MMPR on the coronal image (truncation or cleft sign), (2) vertical high signal at the MMPR on the axial plane, and (3) absence of identifiable MMPR on the sagittal plane, but the meniscus reappears immediately in adjacent images (ghost sign) (Figure 2A–C) [42,43,44]. In addition, medial meniscus extrusion in the coronal plane, a medial displacement of the medial meniscus with respect to the outer border of the medial tibial plateau, can be used to predict MMRTs (Figure 2D) [45,46,47]. Choi et al. reported that a medial meniscus extrusion >3 mm could be associated with the occurrence of an MMRT [47]. The comprehensive use of the abovementioned radiographic findings can facilitate the accurate diagnosis of MMRTs [42,43]. In addition, T2-weighted sequences are considered most useful for identifying MMRTs [41].

Plain radiography or ultrasonography can be used as screening tools for diagnosing MMRTs. Patients with MMRTs reportedly show a relatively decreased medial joint space of the knee on weight-bearing knee radiographs, possibly attributable to osteoarthritis and medial meniscus subluxation [48,49,50]. The peripheral medial joint space-width ratio, which compares the peripheral medial joint space width between the affected and unaffected knees, may be smaller in patients with MMRTs than in those without MMRTs on standing knee radiographs [49]. Similarly, Kodama et al. found that patients with MMRTs showed decreased medial joint space width and increased distance between the medial tibial eminence and medial femoral condyle on weight-bearing posterior-anterior knee radiographs [50]. In the ultrasound examination, medial meniscus extrusion is used to predict the occurrence of MMRTs. Although medial meniscus extrusion is not a pathognomonic sign of MMRTs, parameters indicating the degree of extrusion and the amount of change can be used to predict these tears. On ultrasonographic evaluation, patients with MMRTs have greater medial meniscus extrusion than those without MMRTs [51]. Furthermore, dynamic medial meniscus extrusion, in accordance with changes in weight-bearing conditions, is reduced in patients with MMRTs [52].

## 5. Treatment

Treatment options for MMRTs include non-operative treatment, meniscectomy, surgical repair, and high tibial osteotomy. Although non-operative treatment and partial meniscectomy were the most commonly used treatment options for MMRTs until a decade ago, the frequency of surgical repair has increased as the biomechanical importance of the MMPR has been elucidated [10,25,26,27]. Surgical repair is not appropriate for some cases of MMRTs, and non-operative treatment and meniscectomy remain available in some situations. However, based on the findings of numerous clinical studies, surgical repair has become the treatment of choice for MMRTs.

### 5.1. Non-Operative Treatment

Non-operative treatments include medication, intra-articular injection, and unloader knee braces primarily aimed at relieving the symptoms and also include lifestyle modification and supervised exercise for long-term results [53,54,55,56]. These kinds of treatments can be applied to the following patients who are not suitable for surgical treatments: (1) older patients, (2) those with significant comorbidities, (3) those with advanced knee osteoarthritis (Kellgren-Lawrence grade ≥ 3), and (4) those unwilling to comply with a strict postoperative rehabilitation protocol. In certain groups of patients, symptomatic and functional improvements can be achieved with non-operative treatment [53,54]. However, since meniscal pathology remains unresolved in these cases, clinical improvement is mostly limited to the short term, and degenerative changes in the knee joint are reported to progress [53,54,55,57,58].

### 5.2. Meniscectomy

A torn MMRT tissue can cause impingement between the tibiofemoral joints, leading to pain and mechanical symptoms [12]. Meniscectomy can be a treatment option to relieve these symptoms. Arthroscopic partial meniscectomy for MMRTs can lead to clinical improvement and is reported to show a particularly good prognosis in patients with well aligned non-arthritic knees [12,59,60]. However, the mid-to-long-term clinical results of partial meniscectomy for MMRTs are generally poor, and degenerative changes in the knee joint also progress [61,62]. In addition, arthritic changes in the knee after partial meniscectomy for MMRTs have been reported to be more severe than those after non-operative treatment and surgical repair [58,63,64]. According to a recent study by Yang et al., meniscectomy for MMRTs results in a more significant stress concentration in the tibial cartilage compared with that in untreated MMRTs under dynamic loading conditions [65]. Therefore, owing to its limited clinical benefit, meniscectomy should be performed cautiously in selective cases [29].

### 5.3. Surgical Repair

MMRTs lead to biomechanically detrimental conditions in the knee by creating a functionally meniscal-deficient status, but surgical repair is reported to normalize it [10,25,26,27]. According to a cadaveric study by Padalecki et al., the repair of MMRTs could restore the loading profiles of the knee, being indistinguishable from those conditions with an intact meniscus [26]. Although this finding is based on a time-zero study, it provides a rationale for surgical repair of an MMRT. Along with scientific evidence, unsatisfactory clinical outcomes of meniscectomy have led to increasing interest in performing surgical repair for MMRTs [61,62,63,64].

#### 5.3.1. Indication and Prognostic Factors

The indications for surgical repair of MMRTs are continuously expanding on the basis of numerous studies. Although a consensus has not yet been reached, patients who are suitable for surgical repair of MMRTs are those who meet most of the following conditions: (1) young and physically active, (2) without severe arthritic changes in the knee, (3) without severe varus malalignment of the lower extremity, (4) without high BMI, and (5) willingness to comply with a strict postoperative rehabilitation program [66,67,68,69,70,71,72,73]. In general, older age, high-grade cartilage lesions of the tibiofemoral joint, and varus alignment of the lower extremities are typical poor prognostic factors for surgical repair of MMRTs [66,67,68,69]. Since chronological age differs from biological age, no clear criteria exist for age-related surgical indications. Although 65 years of age is sometimes used as a criterion [38,56,74], several patient characteristics, such as activity level and comorbidities, should be comprehensively considered when deciding on surgery. Arthritic knee changes can be evaluated by plain radiography, MRI, or arthroscopy. It has been reported that radiographic osteoarthritis grade ≥3, according to the Kellgren-Lawrence grade system, and cartilage lesions in the medial tibiofemoral joint of grade ≥3, according to the Outerbridge classification system, are associated with unfavorable prognoses [66,67,68,69,70,75]. Concerning the lower limb alignment, varus >5° is known to be a poor prognostic factor [66,67,68,69]. However, findings from a recent study suggest that the surgical outcomes of patients with a hip-knee-ankle angle between 5° and 10° varus were not inferior to those of patients with a hip-knee-ankle angle <5° varus [74]. Therefore, rather than taking the angle of the lower limb alignment as an absolute criterion, evaluating whether the varus alignment is due to constitutional varus or pathological causes may also be required. High BMI is also reported to be a risk factor for poor outcomes after the surgical repair of MMRTs [70,71]. As in severe varus alignment of the lower extremities, a high BMI may adversely affect the repaired MMRT by increasing its load [75]. In addition, given the relatively low failure load of suture fixation and the limited healing rate of MMRTs [70,72,73], patients should comply with strict postoperative rehabilitation programs to obtain an optimal healing environment [76]. Furthermore, the preoperative symptom duration was found to affect surgical outcomes [38]. A careful patient selection process that considers the aforementioned factors may lead to relatively favorable surgical outcomes. Further research is required to establish optimal surgical indications for the repair of MMRTs.

#### 5.3.2. Surgical Techniques

The repair methods for MMRTs include two techniques: one uses a suture anchor (suture anchor repair), and the other uses a transosseous tunnel (transtibial pull-out repair). Suture anchor repair is performed by placing the suture anchor on the region of an MMRT above the posterior tibial plateau [36,77,78,79,80]. This method has the advantage of reducing the risk of the bungee effect, micromotion between the meniscus-suture complex, and abrasion of the suture material, which may be found in long meniscus-suture constructs, resulting from transtibial pull-out repair [81,82]. However, this surgical method is technically demanding and requires an additional high posteromedial working portal and specialized instruments. It also involves the risk of damaging the cartilage and neurovascular structures [75]. In transtibial pull-out repair, after the suture strands stitched on the torn edge of an MMRT are pulled out through a transosseous tunnel, created from the footprint of the MMRT, to the tibia outer cortex, fixation is made above the anterior tibial cortex. As mentioned, this surgical method may have potential disadvantages, such as micromotion and suture abrasion [81,82]. However, this method is technically less challenging and has a relatively low risk of damage to the vital structures of the knee joint [75]. It also avoids potential complications caused by the loosening of the suture anchor [29]. Therefore, most surgeons use transtibial pullout repair for MMRTs [83]. Although there are few related studies, a paper reported that there are no differences in surgical outcomes between the two surgical techniques for MMRTs [36]. In addition to these two representative techniques, various additional procedures have been used to minimize meniscal extrusion that may persist after surgical repair, including peripheral release, centralization suture, and whip-running suture [84,85,86,87]. However, no well designed clinical study has demonstrated a method that reliably reduces residual meniscus extrusion [88].

Regardless of the surgical method chosen for the repair of MMRTs, it is of utmost importance to ensure that the repair occurs within the anatomical footprint of the MMPR. Unlike anatomical repair, non-anatomical repair cannot restore the loading profile of the knee joint, including the tibiofemoral contact area and pressure [28]. Additionally, complex suture configurations, such as the modified Kessler stitch or modified Mason-Allen stitch, have been reported to show better biomechanical properties than those with simple suture configurations [72,73]. Accordingly, efforts to perform surgical repairs using complex suture configurations at the anatomical footprint with a more familiar surgical technique would be paramount.

#### 5.3.3. Postoperative Rehabilitation

Almost all surgeons apply a rehabilitation program to their patients after the surgical repair of MMRTs. Biomechanically, the load applied to the meniscus generated by weight bearing can affect the repair site [3]. In addition, displacement of the repair construct has been reported after cyclic loading [89]. This displacement may lead to repair loosening and surgical failure. Indeed, the successful healing rate after the surgical repair of MMRTs is approximately 70% [70]. Therefore, well organized postoperative rehabilitation programs are required to provide an optimal healing environment for the repaired MMRT by minimizing the potential effects of harmful stimuli. There are no standardized rehabilitation protocols yet, but most consist of protective strategies, such as limited weight-bearing, range of motion exercises, and brace application [76,90]. The details of the protocols varied greatly among studies. According to a systematic study by Kim et al., range of motion exercises were initiated immediately after surgery or two to three weeks after surgery, and partial weight-bearing was started from one to six weeks after surgery [76]. A knee brace or splint was applied in a fully extended position two to six weeks after surgery [76]. This was followed by progressive muscle-strengthening exercises, and a return to sports is usually allowed six months after surgery [76,90]. Because meniscus-to-bone healing after surgical repair is reported to progress for up to approximately 12 weeks [91], a certain period of protection for the repair construct is necessary to promote healing. However, prolonged protection also carries the risk of developing complications, such as muscle weakness and joint stiffness. Therefore, systematic rehabilitation should be applied under close monitoring, depending on the patient. In addition, accompanying lesions or concomitant surgical procedures should be considered.

#### 5.3.4. Surgical Outcomes

Surgical repair of MMRTs provides both subjective and objective clinical benefits. The literature has consistently reported significant functional improvement, which could persist over the mid- to long-term after surgical repair of MMRTs [83,92,93]. Although surgical repair cannot prevent degenerative changes in the knee joint, the progression of osteoarthritis is less severe compared to that in non-operative treatment or meniscectomy [57,58,94,95]. Additionally, a paper reported that surgical repair is an economically superior treatment approach compared with other treatment modalities [58]. Therefore, although further multidisciplinary studies are required, surgical repair should be recommended for MMRTs, rather than other treatment methods, unless contraindicated [95].

### 5.4. High Tibial Osteotomy

High tibial osteotomy may be a treatment option for MMRTs. Because most MMRTs are degenerative in nature, it is not uncommon for them to be accompanied by cartilage lesions or varus deformities of the lower extremities [96]. As high-grade cartilage lesions in the medial tibiofemoral joint and severe varus malalignment are poor prognostic factors for the surgical repair of MMRTs [66,67,68,69], high tibial osteotomy can be an alternative to the surgical repair of MMRTs for patients with medial compartment osteoarthritis in a varus knee by redistributing the load applied to the knee. High tibial osteotomy has been reported to show favorable clinical outcomes for patients regardless of the healing of MMRTs [97,98]. Furthermore, although it would theoretically be ideal to perform concurrent surgical repair of MMRTs during high tibial osteotomy, it has been reported that there are no clinical benefits in combined surgical procedures compared to isolated high tibial osteotomy [99,100,101].

## 6. Author’s Treatment Strategies for MMRTs

For patients diagnosed with MMRTs, we attempted to perform surgical repair as much as possible, avoiding meniscectomy. If patients with MMRTs do not meet the indications for surgical repair, non-operative treatment or high tibial osteotomy can be suggested by comprehensively considering not only the underlying knee pathology, but also the patient’s characteristics, comorbidities, and willingness for treatment. Our indications for surgical repair of MMRTs are when all of the following are satisfied: (1) not older patients (usually by age 65 years); (2) radiographic osteoarthritis grade ≤2, according to the Kellgren-Lawrence grading system; (3) no pathologic varus alignment of the lower extremity (hip-knee-ankle angle ≤10° and <5° greater compared to the hip-knee-ankle angle of the contralateral lower extremity); and (4) willingness to comply with a strict postoperative rehabilitation program (Figure 3). For the lower limb alignment, we aimed to avoid excluding patients with MMRTs from surgical candidates simply because the hip-knee-ankle angle is >5°, given that constitutional varus is present in a significant portion of the normal population [102,103]. Indeed, we recently found that the short-term clinical outcomes of surgical repair of MMRTs in patients with mild-to-moderate varus alignment were comparable to those in patients with neutral alignment [74]. Treatment plans for cartilage lesions in the medial tibiofemoral joint, which frequently accompany MMRTs, were determined using MRI and arthroscopic findings during surgery. The severity of cartilage lesions was assessed according to the International Cartilage Repair Society (ICRS) grading system [104]. In general, cartilage restoration procedures were not performed when cartilage lesions corresponded to the ICRS grade ≤3a, but were performed for cartilage lesions of the ICRS grade 3b to 3d (if the calcified cartilage layer was exposed). For patients with ICRS grade 4 cartilage lesions, surgical repair of MMRTs, combined with cartilage restoration procedures, was conducted only if they did not want alternative treatment strategies.

The surgical repair of MMRTs is performed through arthroscopic transtibial pull-out repair using a modified reverse Mason-Allen stitch (Figure 4) [38,74]. First, diagnostic arthroscopy is performed via the parapatellar high anterolateral portal, and the accompanying lesions are thoroughly evaluated. When an MMRT is identified, the frayed portion of the torn edge of the MMRT is gently debrided using an arthroscopic shaver, followed by surgical repair. Before the repair process, the medial joint space width is measured using a 5-mm hook on an arthroscopic probe. If the medial gap is considered narrow enough to perform meniscal procedures, a percutaneous pie-crusting release of the superficial medial collateral ligament is performed using a 19-gauge intravenous catheterization needle. This additional procedure reduces the risk of iatrogenic articular cartilage damage and facilitates the surgical procedure without affecting the surgical outcomes and residual valgus laxity of the knee [105]. Subsequently, a crescent-shaped suture hook (Conmed Linvatec) or Knee Scorpion Suture Passer (Arthrex) is inserted into the joint through the anteromedial working portal, followed by making the stitch at approximately 3–5 mm medial to the posterior portion of the torn edge of the MMPR. This process is repeated on the anterior portion of the first stitch, and then a modified reverse Mason-Allen stitch with an ultra-high molecular weight polyethylene (UHMWPE) suture is made (Figure 5) [38,74,106]. We expect this horizontal loop stitch to increase the contact area of the meniscus-to-bone interface by pulling the meniscus, including its peripheral margin, from above. Additionally, using a No. 1 polydioxanone suture (PDS; Ethicon), an overlaid vertical stitch crossing the horizontal loop is created to produce a locking effect [107]. After suturing, an additional anteromedial portal is created. Subsequently, a posteromedial portal is created using the field of view provided by the arthroscope inserted through an additional anteromedial portal. The anatomical footprint of the MMPR is identified under observation through the posteromedial portal, and the bone bed of the footprint is decorticated using an arthroscopic rasp or curette to promote meniscus-to-bone healing. A tibial tunnel guide (Commed Linvatec) is then inserted into the joint through an additional anteromedial portal and placed over the footprint of the MMPR with reference to the remnant stump and posterior cruciate ligament. A tibial transosseous tunnel is created, and a wire loop is inserted into the joint through the tibial tunnel. Next, the sutures stitched to the meniscus are passed through the wire loop and pulled out through the tibial tunnel by pulling the wire loop. Finally, sutures are tied over the EndoButton (Smith & Nephew), which is placed on the anterior tibial cortex just above the pes anserinus. With the knee fully extended, the PDS suture is tied first using a sliding knot while maintaining adequate tension by pulling the UHMWPE suture, and the UHMWPE suture is tied.

Postoperatively, the patients are instructed to adhere to a strict rehabilitation program to ensure optimal conditions for MMRT healing. With the use of crutches and a hinged knee brace in the fully extended position, patients are restricted to toe-touch weight-bearing for four weeks postoperatively, followed by partial weight-bearing (<50% of their body weight) for six weeks. The crutches and brace are discontinued 10 weeks after surgery, and the patients are allowed to have a full weight-bearing gait. Passive range of motion exercises are initiated two weeks postoperatively and gradually increased. Similar to weight bearing, active range of motion exercises are allowed from 10 weeks postoperatively. For muscle strengthening, closed kinetic chain exercises are initiated 10 weeks postoperatively, gradually increasing the exercise intensity [108]. Deep knee flexion should be avoided during daily activities and exercise.

## 7. Summary and Future Direction

MMPR plays a vital biomechanical role in transmitting and distributing the load applied to the meniscus, thereby reducing the impact on the adjacent articular cartilage and preserving the knee joint. Damage to this structure can lead to functional meniscal deficiency, followed by a poor clinical course and rapid progression of osteoarthritis. Most MMRTs are degenerative in nature and usually occur during light daily activities with clinical symptoms, such as painful popping. MMRTs are usually diagnosed based on several characteristic MRI findings. Although there are several treatment options for MMRTs, surgical repair should be prioritized as much as possible to restore the biomechanical function of MMPR, which is supported by the favorable clinical outcomes reported in several clinical studies. However, consensus has yet to be reached regarding the optimal surgical indications, surgical techniques, and rehabilitation protocols for MMRTs. Addressing these issues necessitates further multidisciplinary research. Furthermore, unresolved challenges persist, such as residual meniscus extrusion, the limited healing rate of MMRTs, and the progression of osteoarthritis [38,70,109,110]. Tissue engineering and regenerative medicine may offer potential solutions for these matters. In particular, cell-based tissue regeneration, utilizing various cell sources, is particularly promising as a novel strategy to improve meniscus repair [111,112]. Several aspects in this field also need to be addressed, such as the recruitment and adhesion of cells to the site of meniscus tears, their differentiation into the appropriate cell phenotype, proliferation, and manipulation methods for clinical application [111]. Nonetheless, considering their reparative potential and mounting preclinical evidence, it is expected that most of these issues will be resolved in the near future. Collaboration and joint research with these innovative fields are anticipated to potentially delay the progression of osteoarthritis by overcoming several existing limitations and optimizing the treatment strategies for MMRTs.

## Figures and Tables

**Figure 1 medicina-59-01181-f001:**
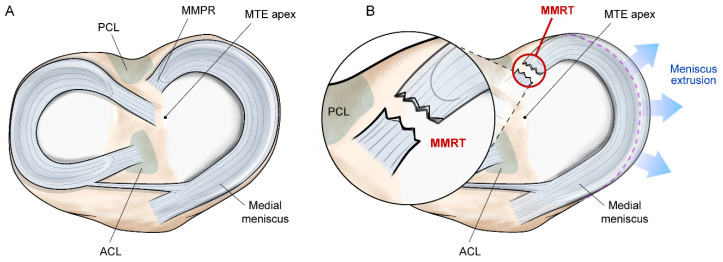
(**A**) An illustration showing the medial meniscus posterior root and its surrounding anatomical structures. (**B**) The occurrence of a medial meniscus posterior root tear leads to the loss of hoop tension in the meniscus, resulting in meniscus extrusion. PCL: posterior cruciate ligament; MMPR: medial meniscus posterior root; MTE: medial tibial eminence; MMRT: medial meniscus posterior root tear; ACL: anterior cruciate ligament.

**Figure 2 medicina-59-01181-f002:**
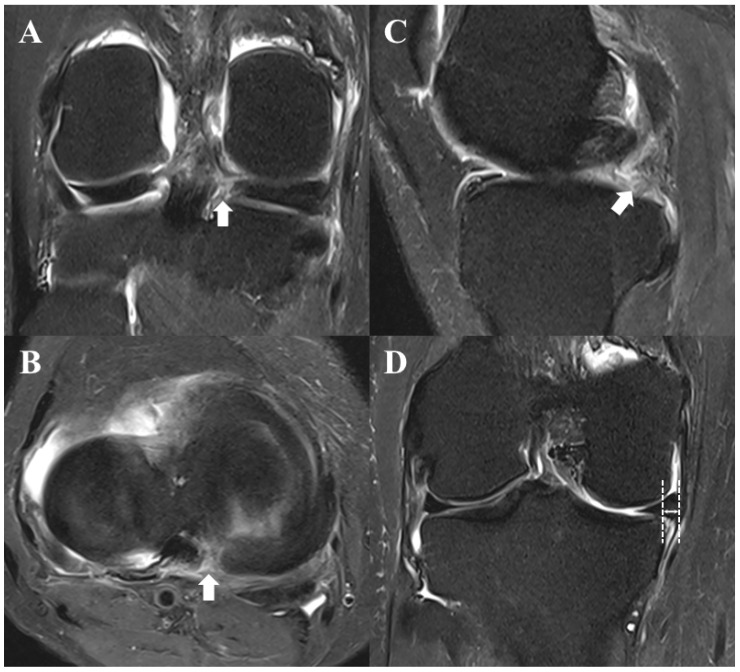
Characteristic magnetic resonance imaging findings, suggestive of a medial meniscus posterior root tear. (**A**) Vertical high signal on the coronal image (truncation or cleft sign; white arrow). (**B**) Vertical high signal on the axial image (white arrow). (**C**) Loss of identifiable medial meniscus posterior root on the sagittal image (ghost sign; white arrow). (**D**) Meniscus extrusion on the coronal image (the length of the white arrow).

**Figure 3 medicina-59-01181-f003:**
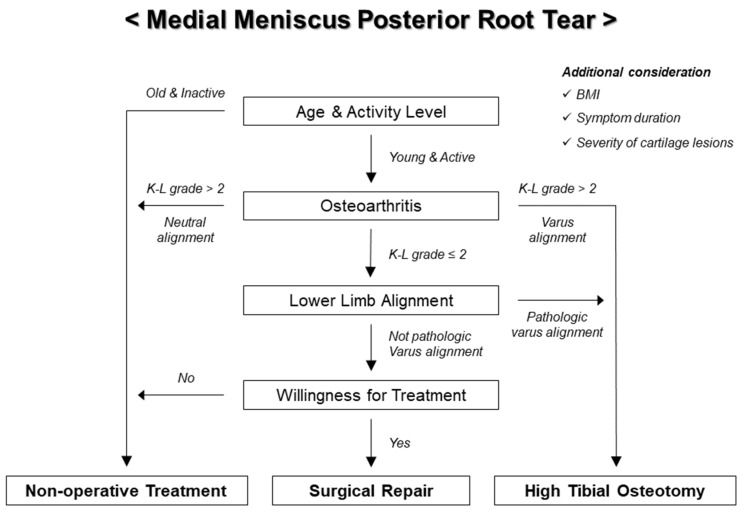
Flowchart of authors’ treatment strategies for medial meniscus posterior root tear. K-L: Kellgren-Lawrence.

**Figure 4 medicina-59-01181-f004:**
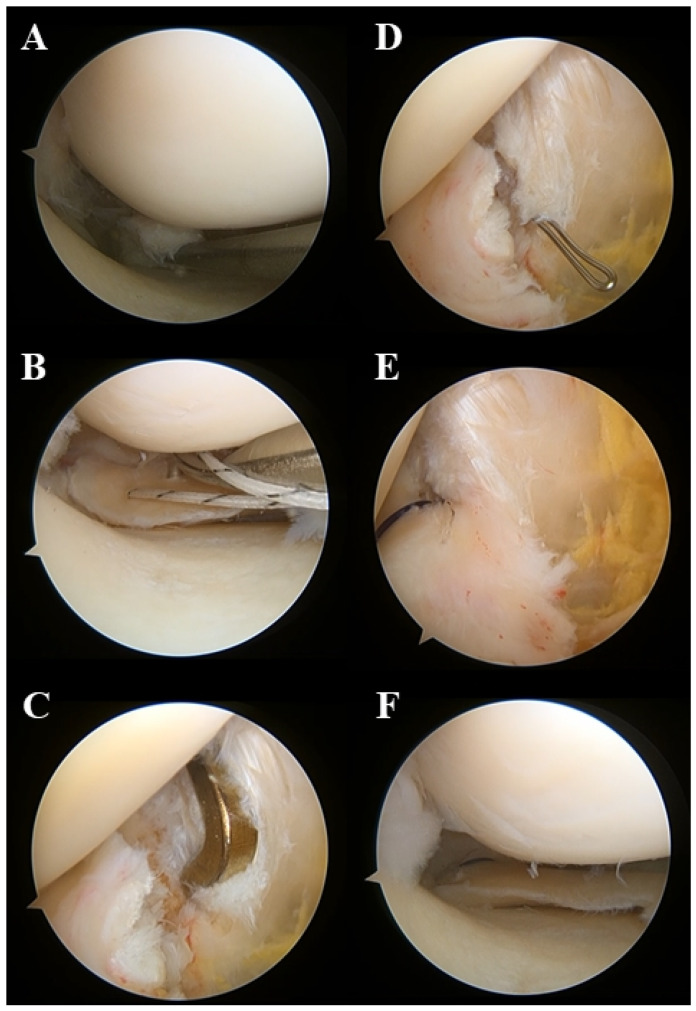
The process of arthroscopic transtibial pull-out repair of a medial meniscus posterior root tear (MMRT). (**A**,**B**) MMRT is confirmed, and a modified reverse Mason-Allen stitch is made (view from the parapatellar high anterolateral portal). (**C**,**D**) After identification of the anatomical footprint of the medial meniscus posterior root and decorticating the bone bed, a transosseous tibial tunnel is created, and a wire loop is inserted into the joint (view from the posteromedial portal). (**E**) Suture strands are pulled out through a tibial tunnel (view from the posteromedial portal) and (**F**) tied (view from the parapatellar high anterolateral portal). Reprinted with permission from Ref. [74]. Copyright 2021, copyright with permission from Moon and Kim.

**Figure 5 medicina-59-01181-f005:**
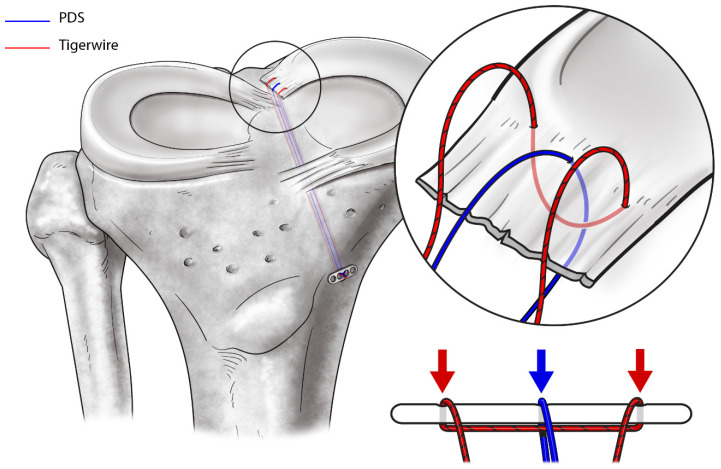
Pictorial illustration of the transtibial pull-out repair of a medial meniscus posterior root tear using a modified reverse Mason-Allen stitch. The stitch consists of the horizontal loop stitch (red thread; ultra-high molecular weight polyethylene suture [Tigerwire suture]) and the overlaid simple vertical stitch (blue thread; No. 1 PDS). Reprinted with permission from Ref. [38]. Copyright 2020, copyright with permission from Moon and Kim.

## Data Availability

Not applicable.

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
