# Peer review of "Medial Meniscus Posterior Root Tear: How Far Have We Come and What Remains?"

_medicina, 2023, doi:10.3390/medicina59071181_

Round 1

Reviewer 1 Report

The Manuscript "Medial Meniscus Posterior Root Tear: How far have we come 2 and what remains?" is a through review in the said area and presents a comprehensive report on the developments in the field.

Introduction: The introduction is too short. It should address the necessity of another review in this area, when several review articles in the similar area are currently available. This is missing. Suggest to include this.

2.1 Anatomy: Suggest to include a figure for easy understanding of the anatomy for wide range of readers. 

3. Biomechanics: Suggest that the load transmission route and the tensions involved should be shown diagramatically. The Mechanics indicating the free body diagram will give better clarity.

4. Clinical presentation: This section contains references which are very old. Try to include maximum references from recent 2-3 years.

5. Treatment: In this section show the various treatment methods using a flow diagram.

7. Summary: The important part of any review paper is the conclusions of the workdone so far and the future direction it gives. The authors need to give more in this direction. Also rename the title of this section as Summary & future direction/scope.

The general standard of English language used is good. Please check for occassional sentence construct errors and grammar.

Author Response

We are grateful to have the opportunity to improve this study. We revised the manuscript based on the given recommendations. To aid the Reviewers and Editors in evaluating the revised manuscript, all new text or the revised parts are highlighted in blue letters in the revised manuscript. Our responses to the Reviewers' comments are provided below.

<Reviewer 1>

The Manuscript "Medial Meniscus Posterior Root Tear: How far have we come and what remains?" is a through review in the said area and presents a comprehensive report on the developments in the field.

Introduction: The introduction is too short. It should address the necessity of another review in this area, when several review articles in the similar area are currently available. This is missing. Suggest to include this.

> Thank you for your comments. As recommended, we revised the Introduction section by describing the need for this study. (Revised Line: 61-63) Since this is a review article, additional descriptions other than mentioning the necessity of this study were not made to prevent unnecessary repetition of the provision of information.

2.1 Anatomy: Suggest to include a figure for easy understanding of the anatomy for wide range of readers. 

> Thank you for your suggestion. We provided relevant Figures showing its anatomy to facilitate readers' understanding. (Figure 1A) Furthermore, we also presented an additional Figure that illustrates meniscus tears and the resulting changes in the meniscus extrusion. (Figure 1B)

3. Biomechanics: Suggest that the load transmission route and the tensions involved should be shown diagramatically. The Mechanics indicating the free body diagram will give better clarity.

> Thank you for your suggestion. As recommended, we expressed this through a relevant Figure. (Figure 1B)

4. Clinical presentation: This section contains references which are very old. Try to include maximum references from recent 2-3 years.

> Thank you for your comments. We totally agree with the reviewer's comments. However, citing only recent papers leads to including similar review articles rather than providing the information of the original articles. Since a review article should also provide information on the original paper that first introduced specific findings, we think it is inevitable to cite somewhat older papers.

5. Treatment:In this section show the various treatment methods using a flow diagram.

> Thank you for your suggestion. We agree that a flowchart of the various treatment options for medial meniscus posterior root tears is required. However, since treatment strategies are continuously expanding and consensus has yet to be reached, we presented a flowchart of our treatment strategies as an alternative. (Figure 3)

7. Summary: The important part of any review paper is the conclusions of the workdone so far and the future direction it gives. The authors need to give more in this direction. Also rename the title of this section as Summary & future direction/scope.

> Thank you for your comments. We supplemented this section and renamed the title to reflect the reviewers' comments. (Revised Line: 432, 441-455)

Reviewer 2 Report

Interesting article, focused on a very debated topic. I have somensuggestions to improve the quality of the manuscript:

-enlarge the introduction section

- the aim of the study should be better specified

- the summary section has to be replaced by “conclusion”, which should resume the findings of the study

Language is adequate

Author Response

We are grateful to have the opportunity to improve this study. We revised the manuscript based on the given recommendations. To aid the Reviewers and Editors in evaluating the revised manuscript, all new text or the revised parts are highlighted in blue letters in the revised manuscript. Our responses to the Reviewers' comments are provided below.

<Reviewer 2>

Interesting article, focused on a very debated topic. I have somensuggestions to improve the quality of the manuscript:

-enlarge the introduction section

> Thank you for your comments. We supplemented the Introduction section by describing the need for this study. (Revised Line: 61-63) Since this is a review article, additional descriptions other than mentioning the necessity of this study were not made to prevent unnecessary repetition of the provision of information.

- the aim of the study should be better specified

> Thank you for your comments. We revised it as you recommended. (Revised Line: 64-66)

- the summary section has to be replaced by “conclusion”, which should resume the findings of the study

> Thank you for your comments. We supplemented this section and renamed the title to reflect the reviewers' comments. (Revised Line: 432)
